# High-harmonic generation in metallic titanium nitride

A. Korobenko [1✉], S. Saha [2], A. T. K. Godfrey[1], M. Gertsvolf[3], A. Yu. Naumov [1], D. M. Villeneuve [1], A. Boltasseva [2], V. M. Shalaev [2] & P. B. Corkum[1]

High-harmonic generation is a cornerstone of nonlinear optics. It has been demonstrated in dielectrics, semiconductors, semi-metals, plasmas, and gases, but, until now, not in metals. Here we report high harmonics of 800-nm-wavelength light irradiating metallic titanium nitride film. Titanium nitride is a refractory metal known for its high melting temperature and large laser damage threshold. We show that it can withstand few-cycle light pulses with peak intensities as high as 13 TW/cm$^2$, enabling high-harmonics generation up to photon energies of 11 eV. We measure the emitted vacuum ultraviolet radiation as a function of the crystal orientation with respect to the laser polarization and show that it is consistent with the anisotropic conduction band structure of titanium nitride. The generation of high harmonics from metals opens a link between solid and plasma harmonics. In addition, titanium nitride is a promising material for refractory plasmonic devices and could enable compact vacuum ultraviolet frequency combs.

[1] Joint Attosecond Science Laboratory, National Research Council of Canada and University of Ottawa, Ottawa, ON, Canada. [2] Purdue University, School of Electrical & Computer Engineering and Birck Nanotechnology Center, West Lafayette, IN, USA. [3] National Research Council Canada, Ottawa, ON, Canada. ✉email: akoroben@uottawa.ca

When intense light irradiates a transparent material, harmonics are generated by the bound electrons or laser-generated free electrons[1–9]. The former is the realm of perturbative nonlinear optics while the latter are responsible for extreme nonlinear optics. Free electron related harmonics are primarily due to newly created free electrons that either recombine after a brief interval in the continuum (inter-band), or after creation, move non-harmonically on the complex bands of the material (intraband). Experiments indicate that, for near-infrared radiation, pre-existing free electrons are not a significant source[10].

In contrast, when normally incident light irradiates a plasma, the high density of free electrons keeps the light out of the material by reflecting it. The phase of the reflected light from a dense plasma is such that it forms a standing wave with a node at the plasma surface. High harmonics from plasmas, observed in many experiments, arise from p-polarized light where electrons are extracted from the surface and the surface discontinuity plays a critical role. A metal, with its high density of electrons, shares many characteristics of plasmas, but the lattice, the resulting band structure and band filling, cannot be ignored.

In this paper, we experimentally study the damage threshold of the epitaxial films of the refractory metal, titanium nitride (TiN). We show that, although lower than expected based on the lattice melting, thermal transport, and light absorption in the material, the damage threshold is still high[11–13], enabling us to observe high harmonics. We find harmonics of 800 nm light reaching 11 eV with brightness comparable to those from magnesium oxide (MgO), a high melting point dielectric, irradiated with the same intensity. Thus, metals can produce high harmonics. We propose that they will occur universally in hard-to-damage bulk metals irradiated with few-cycle pulses.

Because the motion of the conduction electrons is responsible for the plasma response in metals, we develop a simple model, considering the oscillation of the Fermi sea of the laser-driven electrons in a single conduction band of TiN. Extracting the band structure from density functional theory calculations, we use this model to qualitatively predict the angle dependence of the anharmonic motion as the laser polarization is rotated with respect to the lattice structure of the solid. The agreement between the prediction and experiment suggests that the average response of the electrons on the TiN conduction band is an important component of a complete theory.

## Results

**Damage threshold.** Figure 1 shows the layout of the optical setup. To determine the damage threshold, we block the laser beam and adjust its peak intensity with a wire grid polarizer pair. Once the power of the beam is established, it is unblocked, irradiating the film with 60,000 laser pulses. The sample, 200 nm-thick TiN film, epitaxially grown on MgO substrate (see the "Methods" section for details on sample preparation), is then translated by 100 μm to a new spot, and the procedure repeated with a different pulse intensity. After scanning a range of intensities, we removed the sample from the vacuum chamber and inspected it under an optical microscope (Fig. 2a) and an atomic force microscope (AFM) (Fig. 2b).

Comparing the images with the independently measured incident beam profile, we determined the intensity thresholds to be 13 TW/cm$^2$ and 15 TW/cm$^2$ for the TiN modification and ablation, respectively. The damage in pristine MgO was observed at around 50 TW/cm$^2$. Using the two-temperature model approach[14,15] for the photo-induced damage in metal and TiN thermodynamic constants reported previously[16] we estimated the heat deposition depth $x_R = 180$ nm. This corresponds to the

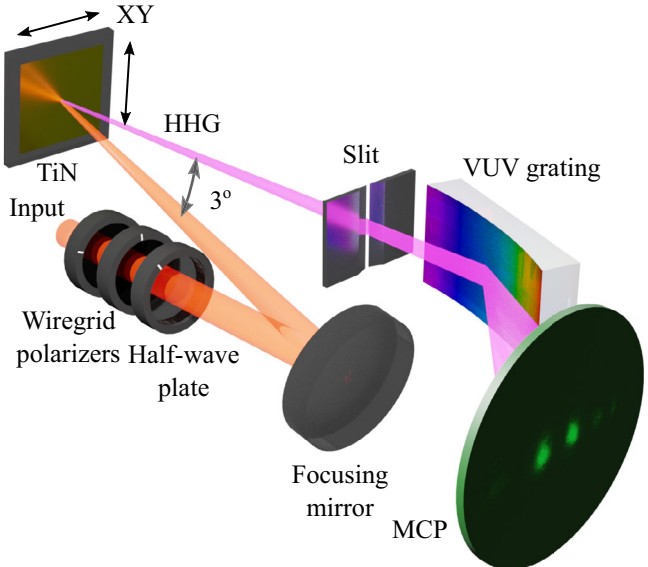

**Fig. 1 Experimental setup.** A 2.3-cycle laser pulse (central wavelength 770 nm) was passed through two wire grid polarizers and a half-wave plate. It was focused with a focusing mirror onto the TiN sample inside a vacuum chamber. The sample was mounted on a motorized XY stage, allowing its translation without realigning the optics. The generated high-harmonics radiation (HHG) passed through a slit, diffracted from a curved VUV grating, and reached the imaging microchannel plate (MCP) detector. The observed VUV spectrum was imaged with a CCD camera.

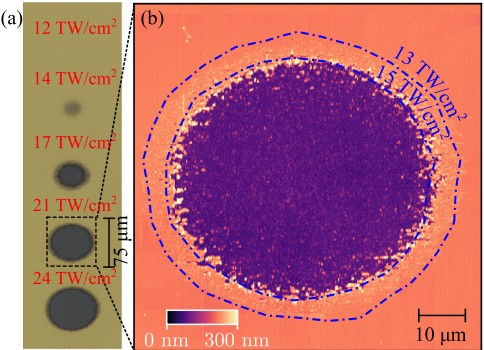

**Fig. 2 Damage threshold measurement. a** Optical microscope image of the irradiated spots on the TiN surface. Numbers 1 through 5 indicate the spots corresponding with the peak field intensities of 12, 13, 17, 21, and 24 TW/cm$^2$ respectively. We observed modification starting from spot #2, and the film appeared stripped, with the underlying MgO exposed at spots #3, #4 and #5. **b** AFM image of spot #4 reveals a ~150 nm-deep crater, surrounded by a halo of swollen TiN material. The bottom of the crater shows a 40-times increase in surface roughness (17 nm RMS), compared to the unmodified region of the sample (0.4 nm RMS), also showing scattered chunks of material with a characteristic size of 100 nm. The two blue dashed-dotted lines are the contour lines of the independently measured incident beam profile, corresponding to the peak intensity of 13 and 15 TW/cm$^2$. These contours set the thresholds for material modification and removal, respectively.

thickness of the TiN layer in which the hot electrons rethermalize with the lattice. The melting temperature of 3,203 K in this layer is achieved at an absorbed fluence of 0.23 J/cm$^2$, which is more than an order of magnitude higher than the experimental threshold fluence of 0.021 J/cm$^2$.

Surprisingly enough, even if we assume that the electrons thermalize with the lattice instantaneously, in which case the heat

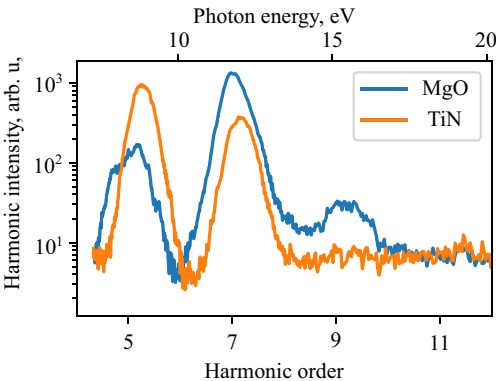

**Fig. 3 High harmonic spectra.** Both the TiN (orange line) and bare MgO substrate (blue line) spectra were taken at incident laser peak intensity of 12 TW/cm$^2$.

deposition length is determined by the (spectral-averaged) TiN absorption length $x_{abs} = 33$ nm, we still get an overestimated damage threshold fluence of 0.043 J/cm$^2$. This suggests non-thermal damage, such as hot-electron blast force[17,18]. However, further study is required to confirm this hypothesis.

**High harmonic generation.** Despite being lower than predicted by a two-temperature model, TiN was still able to withstand an order of magnitude higher incident energy than gold for a similar laser pulse[19]. In addition to this, its relatively low reflection coefficient of 85% allowed us to reach high enough intensity inside the films to observe high harmonics.

The harmonic radiation was emitted in the specular direction to the impinging beam. We collected it in a vacuum ultraviolet (VUV) spectrometer (Fig. 1). With the laser polarization along the [100] crystal direction, we set the laser peak intensity to 12 TW/cm$^2$ and recorded the resulting VUV spectrum, shown in Fig. 3 with an orange line. We calculate the spectral-averaged transmission of our 200 nm-thick film to be $10^{-4}$, eliminating the possible effect of underlying substrate. Harmonic orders HH5 and HH7 (8.4 and 11.8 eV photon energy, respectively) were observed at the intensities below the TiN damage threshold. They were similar in intensity to the reference harmonics from MgO (measured under the same conditions) (Fig. 3, blue line). In addition to HH5 and HH7, harmonic HH9 was also observed from MgO at the intensity range from 10 TW/cm$^2$ to 15 TW/cm$^2$.

Keeping the polarization direction fixed along the [100] crystal direction, we collected a set of spectra, varying the laser pulse attenuation with a wire grid polarizer. Figure 4 summarizes the intensity dependence of the integrated harmonic yield. HH5 and HH7 seem to follow the power laws $I^5$ and $I^7$ (dashed red and magenta lines in Fig. 4), respectively, as a function of the laser intensity $I$. At the intensity of 13 TW/cm$^2$, marked with the green arrow, the monotonic increase of the TiN harmonics gives way to a decrease as material modification occurs. At intensities greater than 15 TW/cm$^2$, marked with the red arrow, the laser radiation ablates the TiN film, revealing the underlying substrate. As a result, the signal at this intensity is dominated by harmonics generated from the MgO under the thinned-out and stripped TiN film at the bottom of the damage crater, and the HH7 curve is following the seventh harmonic intensity scaling we observe in bare MgO (attenuated due to partial absorption in the leftover TiN). The same effect is not observed for HH5 since the latter is too weak in MgO in the studied intensity range to overtake the harmonics emitted by the remaining TiN.

In semiconductors and dielectrics the main high harmonic emission mechanism is interband transitions in which coherent

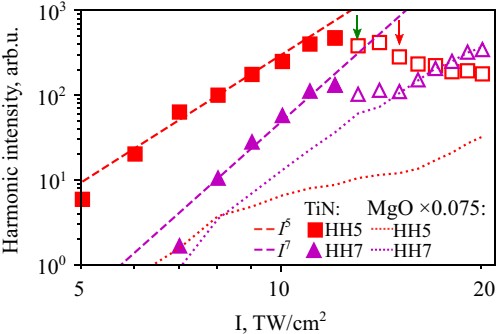

**Fig. 4 Intensity scaling of the harmonics.** Spectrally integrated intensity of HH5 (squares) and HH7 (triangles), measured as a function of input laser intensity at a constant polarization along the [100] crystallographic direction. Empty markers correspond to intensities above the damage threshold, emphasized by the green arrow. Dashed lines are the power laws $I^5$ (red) and $I^7$ (magenta). The dotted lines are the reference MgO harmonics measurements, scaled by a factor of 0.075. At the laser intensities of 15 TW/cm$^2$, marked with the red arrow, and higher, when we observe ablation of TiN film, the HH7 intensity behaves similarly to the MgO, suggesting the latter to be the source of the signal above damage.

electron-hole pairs, produced and driven by a strong laser field, recombine releasing their energy in form of UV photons[3]. In many transparent crystals, including MgO, this recollision process dominates over a co-existing intraband mechanism, stemming from the motion of the electrons in non-parabolic conduction bands[20]. However, as the conduction band population increases (e.g., through optical pre-excitation), the role of the interband processes decreases[10], as the creation of coherent electron-hole pairs is hindered by electrons occupying states near the conduction band minimum.

In contrast, the intraband processes should become more and more important as the free-carrier population increases. (In highly-doped semiconductors, electron-hole creation and recollision at impurity centers still appears to play an important role[21,22], despite the high carrier concentration.) While the photo-carrier density in semiconductors is typically limited at one or a few tens of percent of the conduction band by non-thermal melting[23], metals have much higher electron densities, hinting at the dominant role of the nonlinear conduction current in the HHG process. Analytical theory developed for such current in a 1D 1-band conductor in a tight-binding approximation[24] predicts a power-law intensity scaling for harmonics above the cut-off harmonic number $m_{max} \approx eA_0 a/\hbar \sim 1$, consistent with the observed behavior in Fig. 4. Here $e$ is the elementary charge and $a$ is the lattice constant. Similarly, expanding the field-dependent energy of a 1D single-band conductor in a power series of the crystal momentum $k$, it can be shown that the $m$th spectral component of the induced current has a leading term proportional to $E_0^m$, where $E_0$ is the laser electric field amplitude[25]. The intensity of the $m$-th harmonic would therefore scale as $I^m$, where $I$ is the driving laser intensity, for low enough $I$.

**Harmonics anisotropy.** To gain insight into the origin of the TiN harmonics, we measured their angular dependence. We fixed the intensity and scanned the polarization angle relative to the crystal axes, rotating it with a half-wave plate in the (001) crystallographic plane. The results for input intensity of 11 TW/cm$^2$ are shown in Fig. 5a. Both HH5 and HH7 showed similar anisotropic structure, with the preferable polarization direction along the [100] and symmetrically equivalent crystallographic directions. Comparing the angle dependence of TiN and MgO harmonics,

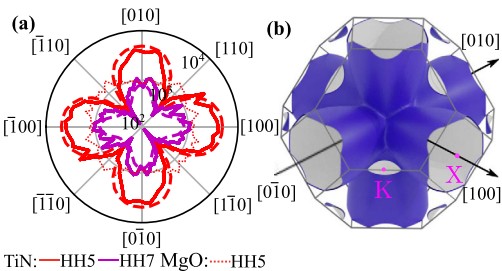

**Fig. 5 Harmonics anisotropy. a** HH5 (solid red) and HH7 (solid magenta) intensity, as a function of the laser polarization angle, at a fixed laser peak intensity of 11 TW/cm². The dashed lines show calculation result. The modeled intensity was scaled up by 20%. For reference, we plot the angular scan of the HH5 intensity from MgO, measured at the same laser peak intensity, with a red dotted line. It demonstrates lower anisotropy, and peaks along [110] and symmetrically equivalent directions. **b** Highly anisotropic Fermi surface of the TiN conduction band. Gray lines represent the edges of the Brillouin zone of the FCC system.

also plotted in Fig. 5a with a dotted red line, identifies their distinctive origins.

We attribute the strong anisotropy of the harmonic yields to the anisotropic conduction band structure of the TiN, resulting in the angular dependence of the screening currents of the conduction electrons. This anisotropy is reflected in TiN's Fermi surface, shown in Fig. 5b. The band consists of 6 valleys, centered at X points of the Brillouin zone, elongated in the $\Gamma X$ direction. This suggests a large difference in the electron dynamics, driven along $\Gamma X$ and $\Gamma K$. However, due to the shape of the conduction band together with its high population, it is not immediately apparent why it would lead to a particular angular dependence plotted in Fig. 5a.

We solved the semiclassical equations of motion to predict the electronic response. We used Density Functional Theory (DFT) to retrieve the electronic bands of TiN. In a dielectric, electrons are mostly excited to the conduction band near a single k-point in the Brillouin zone, where the energy gap is the lowest. 1D calculations following the trajectories of the injected electrons are, therefore, often sufficient to describe high harmonics. For metals, on the other hand, where the electrons in the conduction band start their trajectories from everywhere in the Brillouin zone, full 3D calculations are necessary.

To calculate the harmonic spectra from the band energy $\varepsilon_{\mathbf{k}}$, we use the Boltzmann equation, that, in the absence of scattering or spatial variation electric field of the laser pulse $\mathbf{E}(t)$, has a solution $f_{\mathbf{k}}(t) = f^0_{\mathbf{k}+e\mathbf{A}(t)/\hbar}$. Here, $f_{\mathbf{k}}(t)$ is a time-dependent electron distribution function, $\mathbf{k}$ is the electron crystal momentum, $\mathbf{A}(t) = -\int_{-\infty}^{t} dt' \mathbf{E}(t')$ is the vector potential of the laser pulse, $f^0_{\mathbf{k}} = \frac{1}{\exp(\frac{\varepsilon_{\mathbf{k}}-E_F}{k_B T})+1}$ is the Fermi-Dirac distribution, $E_F$ is the Fermi energy, $k_B$ is the Boltzmann's constant and $T$ is the temperature. We then calculate the current density as:

$$\mathbf{j}(t) = -e \int_{BZ} \frac{d^3\mathbf{k}}{4\pi^3} f_{\mathbf{k}}(t)\mathbf{v}_{\mathbf{k}}, \qquad (1)$$

where $\mathbf{v}_{\mathbf{k}} = \frac{1}{\hbar}\nabla_{\mathbf{k}}\varepsilon_{\mathbf{k}}$ is the electron velocity, $\nabla_{\mathbf{k}}$ is the gradient operator in reciprocal space, and the integration is carried out over a Brillouin zone.

An intense, linearly polarized pulse was numerically propagated through the vacuum/TiN interface, using its measured optical constants (see Methods), to find $\mathbf{A}(t)$ inside. This pulse was then substituted into Eq. (1). to calculate $\mathbf{j}(t)$. We averaged the resulting current density to account for the intensity profile of

the pulse. We then compared the squared amplitude of its Fourier transform with the experiment (Fig. 5a). In agreement with the experimental data, the calculations showed four-fold structure, with a substantial increase of the harmonic yield along [100] and symmetrically equivalent directions.

## Discussion

We found TiN to have a damage threshold an order of magnitude higher than gold, but with evidence of non-thermal damage. The high damage threshold allowed us to observe high harmonics directly from a TiN film, thereby extending the list of high-harmonic generating solids to include metals. The observed spectrum stretched into the technologically important XUV region reaching 11 eV. The next step would be to scale the irradiating intensity to the single-shot damage threshold and beyond.

The measured high harmonics are consistent with intraband harmonics created by conduction band electrons, although we cannot exclude the effect of the higher bands. The harmonic yield is comparable to those generated from the dielectric, MgO, by the same intensity pulse.

Our experiment opens several important technological possibilities. Since TiN is used to make plasmonic devices for on-chip, refractory, and high-power applications[26–32], it will be possible to enhance VUV generation using the field enhancement available with nano-plasmonic antennas[33–35]. One potentially important application is to produce a compact and stable VUV frequency comb. At present the standard way of generating frequency combs is to increase the amplitude of a weak IR frequency comb field in a power-buildup enhancement cavity[36–38], until its intensity is high enough to generate XUV harmonics in a rare gas. We propose to replace the buildup cavity with a TiN nano-plasmonic antenna array and the gas with a dielectric such as MgO[39,40].

Another opportunity is to use TiN as an epsilon-near-zero (ENZ[9]) material to locally enhance the electromagnetic-field and the nonlinear response[9,41,42]. This overcomes the low damage threshold of commonly used transparent conducting oxides such as indium tin oxide (ITO). Since the ENZ wavelength of TiN is around 480 nm[43] and can be adjusted[13,44–46], TiN could pave the way to drastically enhanced nonlinear response.

So far, in our experiments, we remained below the multi-shot modification threshold of TiN. Since the single-shot damage thresholds of TiN should be much higher, we will be able to test harmonic conversion efficiency at a much higher intensity by illuminating the sample with a single laser pulse and collecting the generated harmonics spectra. Furthermore, a single-cycle pulse will allow us to far exceed the single-shot damage threshold and still maintain the crystal structure of TiN. Inertially confined[47] crystalline metals are an uncharted frontier where the many electrons of a metal can be used to efficiently transfer light from the infrared to the VUV.

At higher intensities, the high free carrier concentration in TiN will allow us to study a continuous transition between solid-state high harmonic generation, already linked with gas harmonics, to plasma harmonics, widely studied by the plasma physics community.

## Methods

**Crystal preparation.** A TiN film was deposited using DC magnetron sputtering system (PVD Products) onto a $1 \times 1$ cm² MgO substrate heated at a temperature of 800 °C. A 99.995% pure titanium target of a 2-inch diameter and a DC power of 200 W were used. To ensure high purity of the grown films, the chamber was pumped down to $3 \times 10^{-8}$ Torr before deposition and backfilled to $5 \times 10^{-3}$ Torr with argon during the sputtering process. The throw length of 20 cm ensured a uniform thickness of the grown TiN layer throughout the substrate. After heating, the pressure increased to $1.2 \times 10^{-7}$ Torr. An argon-nitrogen mixture at a rate of 4 sccm/6 sccm was flowed into the chamber. The deposition rate was 2.2 Å/min. The surface quality of the grown films was assessed with an atomic force

microscope. The films are atomically smooth, with a root-mean-square roughness of 0.4 nm. Their optical properties were characterized via spectroscopic ellipsometry at 50 and 70 degrees for wavelengths of 300 nm to 2000 nm and then fitted with a Drude–Lorentz model, with one Drude oscillator modeling the contribution of the free electrons and two Lorentz oscillators modeling the contribution of the bound electrons.

**Optical setup**. We spectrally broadened the 800 nm central wavelength, 1 kHz repetition rate, 1 mJ/pulse energy output of a Ti:Sa amplifier by passing it through an argon-filled hollow-core fiber. Pulses were then recompressed in a chirped-mirror compressor down to 6 fs FWHM duration, as measured with a dispersion scan technique[48].

We focused the beam with a 500 mm focal length concave focusing mirror inside a vacuum chamber onto the TiN (Fig. 1) at a nearly normal incidence angle of 1.5°. The harmonic radiation was emitted from the surface in the specular direction to the incident laser beam, passed through a 300 μm slit of a VUV spectrometer, dispersed by a 300 grooves/mm laminar-type replica diffraction grating (Shimadzu), and an imaging MCP followed by a CCD camera outside the vacuum chamber. We used two wire grid polarizers and a broadband half-wave plate placed outside the chamber to control laser intensity and its polarization. The beam profile at the focal spot was assessed with a CCD camera and found to have a waist radius of 70 μm.

Precise measurement of peak field intensity is difficult in the case of few-cycle pulses. The values reported in this work were calculated from the measured pulse power, beam profile and temporal characteristics of the pulse. The estimated error in pulse intensity was 10%.

**Band structure calculations**. Band structure calculations were performed using GPAW package[49,50], employing a plane-wave basis and PBE exchange-correlation functional, that was found to yield good results in previous DFT studies of TiN[51]. Having performed the calculations on a rough $16 \times 16 \times 16$ $k$-point grid we used Wannier interpolation to interpolate the band energy $\varepsilon_{\mathbf{k}}$ to a denser $256 \times 256 \times 256$ one with wannier90 software[52].

The resulting band structure had three energy branches crossing the Fermi level, consistent with previous studies[51,53]. Two of them had a minimum at the center of the Brillouin zone, Γ, contributing 0.08 and $0.13 \times 10^{28}$ m$^{-3}$ to the conduction band electron density. The third one, whose Fermi surface is shown in Fig. 5b, was highly anisotropic and minimized at the X point. Corresponding to the electron density $5.03 \times 10^{28}$ m$^{-3}$ it was dominant for generating high harmonics.

## Data availability
The datasets generated during and/or analyzed during the current study are available in the figshare repository, https://doi.org/10.6084/m9.figshare.c.5514561.v1.

## Code availability
The code used for data analysis is available in the figshare repository, https://doi.org/10.6084/m9.figshare.c.5514561.v1.

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

## Acknowledgements

The work was funded by US Defense Threat Reduction Agency (DTRA) (HDTRA1-19-1-0026) and the University of Ottawa, NRC Joint Centre for Extreme Photonics; with contributions from the US Air Force Office of Scientific Research (AFOSR) FA9550-16-1-0109, FA9550-18-1-0002, FA9550-20-01-0124 and ONR grant N00014-20-1-2199; Canada Foundation for Innovation; Canada Research Chairs (CRC); and the Natural Sciences and Engineering Research Council of Canada (NSERC). We thank David Crane and Ryan Kroeker for their technical support, and are grateful for fruitful discussions with Andre Staudte, Giulio Vampa, Guilmot Ernotte and Marco Taucer.

## Author contributions

S.S. synthesized and characterized linear properties of the TiN films. A.K. performed and analyzed DT and HHG measurements and carried out numerical calculations. A.T.K.G. conducted AFM characterization. P.B.C. supervised and directed the project. A.K., S.S., A.T.K.G., M.G., A.Y.uN., D.M.V., A.B., V.M.S., P.B.C. contributed to discussing the results and writing the manuscript.

## Competing interests

The authors declare no competing interests.
