## [Peer Review File · Nature Communications]

REVIEWER COMMENTS

Reviewer #1 (Remarks to the Author):

The authors describe high order harmonic laser emission from a metallic sample. This highly nonlinear phenomenon, known since the mid-80s in gas medium, has been observed for the first time in crystalline samples in 2011. Since then, a large range of solid media has been used, from semiconducting to dielectric crystals to 2D materials. However, experiments in metals have so far been prevented by their low damage thresholds.

Using a refractory metal, which has a damage threshold one order of magnitude larger than usual metals, and few cycle laser pulses in order to reduce energy deposition, the authors demonstrate for the first time high order harmonic generation in a metallic thin TiN film. While this result is interesting for the ultrafast community, the results seem to be too specific to this journal's readers. Moreover, the stringent experimental conditions necessary to observe the emission (notably the almost single cycle laser pulses) will limit the applicability of metals as a practical source of short wavelength ultrashort coherent pulses, despite the few examples that the authors give towards the end of the manuscript. In my opinion, the manuscript should maybe be re-submitted to a physics journal.

Additionally, I have some questions and remarks on the manuscript.

1. I think the statement line 84 to 86 is misleading. Indeed, the authors used DFT to calculate the band structure of TiN, but DFT was not used to simulate the HHG per se. Indeed, if I am not mistaken, one would need the time dependent version of DFT for this. The authors should clarify this point.
2. Related to Fig. 2 and the note about intensity estimation in the Methods section, I find puzzling the high accuracy of the intensity values given in the figure caption. How reliable are those numbers?
3. Figure 3, the authors show HHG spectra attributed to TiN and to MgO, which have similar intensities. However, how did the authors make sure that the emission from the first panel is not due to MgO? I could not find the thickness of the TiN film. Moreover, they should give the absorption coefficients of TiN in the wavelength range of interest, in order to rule out emission from MgO as the origin of the first spectrum.
4. Why in fig. 4 are the HHG signal more intense in panel b than in panel a? In addition, there is a discrepancy between the intensity given in the caption of panel b (l179) and in the main text (l192).
5. L 166 to 168 and fig. 4a: it would be useful to see the intensity dependence of MgO HH7 for comparison, as the authors claim that the emission measured at high intensity is coming from MgO. Did they measured the HH signal for higher intensities? If not, claiming that the signal from HH7 is following MgO intensity scaling from only 2 points is not really convincing. They should either show the additional points if they were measured or explain their claim. On the same graph, why is the HHG yield decreasing for H5 at high I? What can explain the different behavior between HH5 and HH7?
One last question in this regards: What is the damage threshold of MgO?
6. L 211-213. Where is this expectation coming from? The authors should explain how they came to this hypothesis. I think they should also mention the work from Wang et al., Nature Comm. 8, 1686 (2017), leading to a situation somehow similar to the one here, i.e. with a large population of electrons in the conduction band. However, they reach a different conclusion, claiming that the main mechanism for HHG is in their case interband electron dynamics. Can the authors discuss decoherence effects that could hamper the HHG process?
7. L 284-285: regarding HHG in MgO, the authors say that "the harmonics appear to have an interband

origin". This was not the subject on this work, thus they should either justify their claim from some data or calculation or cite some reference.

8. L 286-299: While the idea is interesting, it is not really new, and I do not think it belongs to the present study. Indeed, using TiN for plasmonic devices is already known. Moreover, in the current study, HHG takes place inside the TiN. Using plasmonic structure in TiN embedded in another medium to boost HHG from this medium would deserve a dedicated study. To me, adding this unrelated idea looks like a way to attract more citation from future works to this paper, which I do not approve. The application to HHG in ENZ materials is to my mind more appropriate, as it used the same nonlinear medium. On a similar line of thought, I have the feeling that the reference to the inertially confined metals (a notion that is not defined at all) is just a way to increase the citation rate of one of the authors' work (author who does not need this at all).

9. L 322-327: While it is important to show the potential of one's work, pushing the idea too far is somehow counterproductive. The authors showed in their work that at higher intensities, even TiN gets damaged. Using single cycle pulses, the damage threshold would indeed become larger, but only to a limited extent, and certainly not as high as to enable plasma harmonics.

10. L 357: the authors should add a reference to the dispersion scan technique they used.

11. L 376: Can the authors give a quantitative estimate of this uncertainty? This would be useful to know how reliable the 2 thresholds for the 2 damage regimes mentioned in the manuscript, as they are really close one from the other.

12. Some abbreviations are not defined: HHG, HH.

Reviewer #2 (Remarks to the Author):

Korobenko et al. reported the high-harmonic generation in metallic TiN films with a high damage threshold for the first time. The HHG photon energy is up to 11 eV. They also studied the intensity scaling and angular anisotropy of the HHG photons, confirming the intraband mechanism. It is an interesting work that could be published in Nature Communications. I have the following comments:

1)What is the thickness of the TiN film?

2)What is the transmission coefficient of TiN for the 770 nm driving laser? are there contributions from the substrate MgO in the HHG signals in the top panel of Fig. 3?

3)Is the HHG response perturbative or nonperturbative? Two lines I^N should be plotted in Fig. 4(a) for references.

4)Pronin, Bandrauk, and Ovchinnikov published a theoretical paper [PRB 50, 3473 (1994)] about HHG in a conductor. They found that the maximum HHG energy is E_a . My questions are: Does the cutoff energy (7th harmonic) in your experiments agree with this law? Does the cutoff energy change if you use another driving laser wavelength?

XueBin Bian, supporting the transparent peer review.

Reviewer #3 (Remarks to the Author):

The manuscript by Korobenko et al. reports high harmonic generation in a metallic TiN film using linearly-polarized few-cycle 770-nm light pulses with a maximum intensity of 13 TW/cm² (=100 MV/cm) in the reflection configuration. The authors observed high harmonics up to the 7th order in a TiN film and to the 9th in a bulk MgO in the same setup. The intensity scaling and anisotropy of high harmonic yields are investigated and compared with simulation that assumes a field-free band structure and the intraband current induced in the lowest conduction band starting at the X point. Based on the decent agreement of the experiment and simulation, the authors concluded that the origin of efficient high harmonic generation in TiN was the intraband current.

Although the results are interesting in two points – (i) TiN can produce the 5th harmonics more efficiently than MgO, which is already in VUV, and (ii) TiN withstands to a very high intensity up to 12 TW/cm² (=100 MV/cm), the validity of the model is questionable in the following points:

(1) At an intensity of >10 TW/cm², it is not clear whether the field-free band structure can be assumed.
(2) In the scaling data [Fig.4(a)], the HH5 curve is highly deviated from the calculation at I>10 TW/cm², while the HH7 curve deviates at I<8 TW/cm². I suppose that these behaviors are difficult to understand with their intraband picture. If their calculation is correct, HH7 can be observable below 8 TW/cm², for example. Because of the lack of validity in their simulation, their conclusion is not convincing enough.

In addition, I have several questions which the authors should address.

(3) One of the novel points of this paper is that the sample is metal, not semiconductors or insulators. However, this difference is not essential. It is possible to inject high density carriers in semiconductors for example.

(4) As I understand, the authors only used the intraband current in the lowest conduction band. Is there any chance that electrons are launched to higher conduction bands at such a high intensities?

(5) It is helpful to show the energy map (ϵ_k) of the band of interest instead of the Fermi surface shown in Fig. 4(b). With such a figure, the authors can also show the excursion range of electrons around the X point and why HHG is heavily suppressed in some directions.

In summary, although experimental results are interesting and suggesting for possible device applications, the theoretical model does not fully support the experimental observation. Therefore, their conclusion that the intraband current is the source of HHG is not be convincing enough. The associated physics of HHG is not new at all, and I don't see sufficient technological advances either. I thus conclude that the manuscript is not acceptable as it is to Nature Communications.

Dear reviewers and editors,

Thank you for your time and effort put into reviewing our work. Please, find below your comments and questions addressed.

Reviewer #1 (Remarks to the Author):

The authors describe high order harmonic laser emission from a metallic sample. This highly nonlinear phenomenon, known since the mid-80s in gas medium, has been observed for the first time in crystalline samples in 2011. Since then, a large range of solid media has been used, from semiconducting to dielectric crystals to 2D materials. However, experiments in metals have so far been prevented by their low damage thresholds.

Using a refractory metal, which has a damage threshold one order of magnitude larger than usual metals, and few cycle laser pulses in order to reduce energy deposition, the authors demonstrate for the first time high order harmonic generation in a metallic thin TiN film. While this result is interesting for the ultrafast community, the results seem to be too specific to this journal's readers. Moreover, the stringent experimental conditions necessary to observe the emission (notably the almost single cycle laser pulses) will limit the applicability of metals as a practical source of short wavelength ultrashort coherent pulses, despite the few examples that the authors give towards the end of the manuscript. In my opinion, the manuscript should maybe be re-submitted to a physics journal.

Thank you for your review. We would like to take this opportunity to comment on the manuscript appropriateness for the *Nature Communications*.

- (a) We believe that observing high harmonics from metals is of general interest. To make this clear to a reader we have reworked the introduction to the paper to place the manuscript in a broader context than in the first draft. Specifically, we now discuss AMO, dielectrics and semiconductors, and plasma harmonic research.

We have also taken the referee's criticisms to heart and made several important changes:

- (b) We have added an analysis of the damage threshold of TiN to the text. We find that the damage threshold that we measure ($1.3 \times 10^{13} \text{ W/cm}^2$) is much below melting temperature of our 200 nm thick TiN sample.
- (c) To address the perturbative harmonic response of metals, we present a 1D model of harmonic emission from metals.
- (d) We have clarified our role as experimentalists, not theorists. Our experimental measurement of the angle dependence of the harmonics and its relation to the Fermi surface, show that an essential part of any theory will follow the motion of the average metal electron. Pointing to important directions for a full theory is exactly the role of a good experiment.

Additionally, I have some questions and remarks on the manuscript.

1. I think the statement line 84 to 86 is misleading. Indeed, the authors used DFT to calculate the band structure of TiN, but DFT was not used to simulate the HHG per se. Indeed, if I am not mistaken, one would need the time dependent version of DFT for this. The authors should clarify this point.

We agree and have corrected this sentence to

“Our simple model uses density functional theory to calculate the band occupation of the conduction electrons.”

2. Related to Fig. 2 and the note about intensity estimation in the Methods section, I find puzzling the high accuracy of the intensity values given in the figure caption. How reliable are those numbers?

We corrected the values and provided the intensity uncertainty estimate at 10% in the “Optical setup” section under “Methods”.

“The values reported in this work were calculated from the measured pulse power, beam profile and temporal characteristics of the pulse. The estimated error in pulse intensity was 10%.”

3. Figure 3, the authors show HHG spectra attributed to TiN and to MgO, which have similar intensities. However, how did the authors make sure that the emission from the first panel is not due to MgO? I could not find the thickness of the TiN film. Moreover, they should give the absorption coefficients of TiN in the wavelength range of interest, in order to rule out emission from MgO as the origin of the first spectrum.

We added the sample thickness to the manuscript.

“we use an epitaxially grown TiN on MgO film that is 200 nm thick”

We estimated the spectral-averaged transmittivity of the film using the optical constants from the ellipsometer measurements at 0.01% of intensity of our laser beam, which combines the reflectivity of the interfaces as well as the absorption in TiN. From calculations, the intensity of the pump laser radiation reaching MgO (0.01% of pump) is too low to drive any detectable XUV at all studied intensities. Moreover, any XUV generation generated by MgO would still have to penetrate the TiN film to reach the detector. From the TiN XUV absorption data reported before, it shows that less than 10^{-7} of that would get transmitted. We added these estimates to the manuscript to back up our claims.

“We calculate the spectral-averaged transmission of our 200 nm-thick film to be 10^{-4} , eliminating the possible effect of underlying substrate.”

4. Why in fig. 4 are the HHG signal more intense in panel b than in panel a? In addition, there is a discrepancy between the intensity given in the caption of panel b (I179) and in the main text (I192).

We thank the reviewer for pointing out the inconsistency. The discrepancy stems from the polarization sensitivity of the XUV grating that we compensate with an additional factor. We have now scaled both panels to the same arbitrary units. We also fixed the error in the intensity quoted.

“The results for input intensity of 11 TW/cm^2 are shown in Fig. 5 (a).”

5. L 166 to 168 and fig. 4a: it would be useful to see the intensity dependence of MgO HH7 for comparison, as the authors claim that the emission measured at high intensity is coming from MgO. Did they measured the HH signal for higher intensities? If not, claiming that the signal from HH7 is following MgO intensity scaling from only 2 points is not really convincing. They should either show the additional points if they were measured or explain their claim. On the same graph, why is the HHG yield decreasing for H5 at high I? What can explain the different behavior between HH5 and HH7? One last question in this regards: What is the damage threshold of MgO?

The measurements were performed up to the peak intensities of 20 TW/cm^2 , way past the damage threshold of TiN. As the film get damaged, the XUV radiation generated in TiN decreases in yield, but can give way to the emission from the stripped MgO surface below. This emission is still lower then the pristine MgO surface generates due to the scattering and absorption of the film leftover on the surface.

We believe that the reason why this was only observed for HH7 and not HH5 is that the latter is much weaker in MgO and did not reach a high enough value to exceed the signal from the TiN leftover. We added the curves for the MgO harmonics to the plot (Fig.3 and Fig. 4).

The damage threshold for MgO was observed at around 50 TW/cm^2 . We added this clarification information to the text.

“The damage in pristine MgO was observed at around 50 TW/cm^2 .”

6. L 211-213. Where is this expectation coming from? The authors should explain how they came to this hypothesis. I think they should also mention the work from Wang et al., Nature Comm. 8, 1686 (2017), leading to a situation somehow similar to the one here, i.e. with a large population of electrons in the conduction band. However, they reach a different conclusion, claiming that the main mechanism for HHG is in their case

interband electron dynamics. Can the authors discuss decoherence effects that could hamper the HHG process?

The interband HHG takes place in three steps. First, the electron-hole pairs are strong-field excited across the band gap, then the electron and hole are separated by the oscillating field, which then recollides them again with the excess energy released as a phonon. The excitation happens near the high-symmetry points of the band, where the energy difference is minimal. Injecting even a small number of thermalized carriers into the conduction band quickly hinders the electron-hole pair creation, as they occupy the states to which the strong-field transitions occur. On the other hand, the intraband HHG benefits from the free carriers. This is consistent with the findings of Wang et al. (see e.g. Fig 3 and its discussion in the text), with the important difference being that the carrier densities of a metal are unachievable in semiconductors through photoexcitation due to the damage [Rousse et al. Nature **410**, 65]. We added this reference and clarifying discussion to the text.

“However, as the conduction band population increases (e.g., through optical pre-excitation), the role of the interband processes decreases (13), as the creation of coherent electron-hole pairs is hindered by electrons occupying states near the conduction band minimum.

In contrast, the intraband processes should become more and more important as the free-carrier population increases. (In highly-doped semiconductors, electron-hole creation and recollision at impurity centers still appears to play an important role (21, 22), despite the high carrier concentration.)”

7. L 284-285: regarding HHG in MgO, the authors say that “the harmonics appear to have an interband origin”. This was not the subject on this work, thus they should either justify their claim from some data or calculation or cite some reference.

We added a reference to justify this claim.

“In many transparent crystals, including MgO, this recollision process dominates over a co-existing intraband mechanism, stemming from the motion of the electrons in non-parabolic conduction bands (20).”

8. L 286-299: While the idea is interesting, it is not really new, and I do not think it belongs to the present study. Indeed, using TiN for plasmonic devices is already known. Moreover, in the current study, HHG takes place inside the TiN. Using plasmonic structure in TiN embedded in another medium to boost HHG from this medium would deserve a dedicated study. To me, adding this unrelated idea looks like a way to attract more citation from future works to this paper, which I do not approve. The application to HHG in ENZ materials is to my mind more appropriate, as it used the same nonlinear medium. On a similar line of thought, I have the feeling that the reference to the inertially confined metals (a notion that is not defined at all) is just a way to increase the citation rate of one of the authors’ work (author who does not need this at all).

We see the referee's point, but respectfully disagree. We believe it is important to provide an outlook for the future research. We are not claiming that we invented the plasmonic-enhanced harmonics in this work but do think our findings can provide a new perspective on it, where the material works both as a plasmonic device as well as HHG medium. In fact, this study was in part motivated by the previous work from our group on the plasmon-enhanced HHG [Vampa et al., Nat. Phys., **13**, 659], where the disadvantages of gold as a plasmonic material was discussed, and TiN was mentioned as a potential substitute.

9. L 322-327: While it is important to show the potential of one's work, pushing the idea too far is somehow counterproductive. The authors showed in their work that at higher intensities, even TiN gets damaged. Using single cycle pulses, the damage threshold would indeed become larger, but only to a limited extent, and certainly not as high as to enable plasma harmonics.

We do agree that probably no material can withstand the intensities used in plasma harmonics. However, the lattice damage, determined by the electron-phonon coupling and further slowed-down by lattice inertia, occur on timescales much longer than a duration of a single few-cycle pulse. As a result, employing refractory metal antennae, and plasmonic structures where the field enhancements might lower the required intensities, can be both viable means to enable plasma harmonics, and warrant exploration.

10. L 357: the authors should add a reference to the dispersion scan technique they used.

Added the reference [45] M. Miranda, T. Fordell, C. Arnold, A. L'Huillier, H. Crespo, Simultaneous compression and characterization of ultrashort laser pulses using chirped mirrors and glass wedges. Opt. Express. 20, 688–697 (2012).

11. L 376: Can the authors give a quantitative estimate of this uncertainty? This would be useful to know how reliable the 2 thresholds for the 2 damage regimes mentioned in the manuscript, as they are really close one from the other.

We corrected the values and added an estimate for intensity uncertainty.

"The values reported in this work were calculated from the measured pulse power, beam profile and temporal characteristics of the pulse. The estimated error in pulse intensity was 10%."

12. Some abbreviations are not defined: HHG, HH.

Added definitions.

Reviewer #2 (Remarks to the Author):

Thank you, Prof. Bian, for your review.

Korobenko et al. reported the high-harmonic generation in metallic TiN films with a high damage threshold for the first time. The HHG photon energy is up to 11 eV. They also studied the intensity scaling and angular anisotropy of the HHG photons, confirming the intraband mechanism. It is an interesting work that could be published in Nature Communications. I have the following comments:

1)What is the thickness of the TiN film?

The thickness of 200 nm was added to the text.

“we use an epitaxially grown TiN on MgO substrate that is 200 nm thick”

2)What is the transmission coefficient of TiN for the 770 nm driving laser? are there contributions from the substrate MgO in the HHG signals in the top panel of Fig. 3?

We calculate the spectrum-averaged transmission to be $1e-4$, which eliminates the possibility of the substrate effect. We added this estimate to the text.

“We calculate the spectral-averaged transmission of our 200 nm-thick film to be 10^{-4} , eliminating the possible effect of underlying substrate.”

3)Is the HHG response perturbative or nonperturbative? Two lines I^N should be plotted in Fig. 4(a) for references.

We find the experimental curves following I^N dependence quite closely (Fig. 4). We added those curves to the plots, as requested, and added more discussion on the observed dependencies.

“Indeed, as shown in Methods, an oscillating electric field $E(t)$ induces current density

$$j(t) = \sum_{m=1,3,5,\dots} \cos m\omega t \sum_{n=m,m+2,\dots} B_{nm} A_0^n$$

in a 1D metal... This expression suggests that the amplitude of the m -th odd harmonic of the nonlinear current has a leading term proportional to A_0^m . The intensity of the m -th harmonic would therefore scale as I^m , where I is the driving laser intensity, for low enough I .”

4)Pronin, Bandrauk, and Ovchinnikov published a theoretical paper [PRB 50, 3473 (1994)] about HHG in a conductor. They found that the maximum HHG energy is E_a . My questions are: Does the cutoff energy(7th harmonic) in your experiments agree with this law? Does the cutoff energy change if you use another driving laser wavelength?

We thank the reviewer for providing this reference. The strongest fields used in our experiment, $E \sim 4e9$ V/m, is predicted to correspond to a cut-off harmonic number of a mere 1.2. The authors also predict the beyond cutoff harmonics to scale as I^N , which agrees well with the experimental observation.

"This is also consistent with the analytical model, developed for a 1D 1-band conductor (24), that predicts a power-law intensity scaling for harmonics above the cut-off harmonic number $m_{max} \approx eA_0 a / \hbar \sim 1.$ "

XueBin Bian, supporting the transparent peer review.

Reviewer #3 (Remarks to the Author):

Thank you for your review.

The manuscript by Korobenko et al. reports high harmonic generation in a metallic TiN film using linearly-polarized few-cycle 770-nm light pulses with an maximum intensity of 13 TW/cm² (=100 MV/cm) in the reflection configuration. The authors observed high harmonics up to the 7th order in a TiN film and to the 9th in a bulk MgO in the same setup. The intensity scaling and anisotropy of high harmonic yields are investigated and compared with simulation that assumes a field-free band structure and the intraband current induced in the lowest conduction band starting at the X point. Based on the decent agreement of the experiment and simulation, the authors concluded that the origin of efficient high harmonic generation in TiN was the intraband current.

Although the results are interesting in two points – (i) TiN can produce the 5th harmonics more efficiently than MgO, which is already in VUV, and (ii) TiN withstands to a very high intensity up to 12 TW/cm² (=100 MV/cm), the validity of the model is questionable in the following points:

(1) At an intensity of >10 TW/cm², it is not clear whether the field-free band structure can be assumed.

The calculations that we provide were not expected to provide the detailed quantitative description of the observed effect, but rather aim at their qualitative prediction. In the new paper revision, we emphasized the experimental nature of our work.

Having said that, while the referee is right, and the band structure is probably modified in such intense fields, field-free bands are routinely successfully used to describe HHG in strong-field-driven solids, e.g.: Luu, et al. Nature, **521** 498, You, et al. Opt. Lett., **42** 1816, and we expect the theory to hold true for our experimental conditions.

(2) In the scaling data [Fig.4(a)], the HH5 curve is highly deviated from the calculation at $I > 10 \text{ TW/cm}^2$, while the HH7 curve deviates at $I < 8 \text{ TW/cm}^2$. I suppose that these behaviors are difficult to understand with their intraband picture. If their calculation is correct, HH7 can be observable below 8 TW/cm^2 , for example. Because of the lack of validity in their simulation, their conclusion is not convincing enough.

We thank the reviewer for this valuable comment. Due to the simplicity of our model, we do agree that the predicted intensity scaling law might not agree convincingly well with the experiment. In the new revision of the manuscript, we are providing a comparison with simple power laws, instead, which are predicted for 1D conductor.

In addition, I have several questions which the authors should address.

(3) One of the novel points of this paper is that the sample is metal, not semiconductors or insulators. However, this difference is not essential. It is possible to inject high density carriers in semiconductors for example.

While it might be possible to photoexcite a substantial free-electron density, it remains well below the electron density in a metal, limited by non-thermal melting (e.g. Rousse et al., Nature **410**, 65). In fact, the published work on the effect of photo-injection in semiconductors on HHG, showed that the detrimental effect of photo-injections, as the injected electrons, thermalizing at the bottom of the conduction band and occupying the states needed for coherent electron-hole pairs creation, hindered the main mechanism of harmonic generation, interband emission (Wang et al., Nat. Commun. **8**, 1686). We added discussion of these points to the text:

“However, as the conduction band population increases (e.g., through optical pre-excitation), the role of the interband processes decreases (13), as the creation of coherent electron-hole pairs is hindered by electrons occupying states near the conduction band minimum.

In contrast, the intraband processes should become more and more important as the free-carrier population increases. (In highly-doped semiconductors, electron-hole creation and recollision at impurity centers still appears to play an important role (21, 22), despite the high carrier concentration.)”

(4) As I understand, the authors only used the intraband current in the lowest conduction band. Is there any chance that electrons are launched to higher conduction bands at such a high intensities?

We cannot currently exclude the effect of higher bands. Our model seems to not contradict the intraband current picture, but further experiments are required to exclude this effect. We added the following sentence to the text

“The measured high harmonics are consistent with intraband harmonics created by conduction band electrons, although we cannot exclude the effect of the higher bands.”

(5) It is helpful to show the energy map (epsilon_k) of the band of interest instead of the Fermi surface shown in Fig. 4(b). With such a figure, the authors can also show the excursion range of electrons around the X point and why HHG is heavily suppressed in some directions.

It is not apparent from the band structure, why the HHG is suppressed along some directions. The reason is that the conduction electrons' starting momenta occupy essentially half of the Brillouin Zone, and during their excursion they experience substantially different dispersion curves. This is why we abstained from drawing one particular dispersion curve, as the harmonics are a result of all electrons trajectories, and drawing just one would not capture the current anisotropy. We added the following clarification to the text:

“However, due to the involved shape of the conduction band together with its high population, it is not immediately apparent why it would lead to a particular angular dependence plotted in Fig. 5 (a).”

In summary, although experimental results are interesting and suggesting for possible device applications, the theoretical model does not fully support the experimental observation. Therefore, their conclusion that the intraband current is the source of HHG is not be convincing enough. The associated physics of HHG is not new at all, and I don't see sufficient technological advances either. I thus conclude that the manuscript is not acceptable as it is to Nature Communications.

We would like to take this opportunity to comment on the manuscript appropriateness for the Nature Comm. We believe that observing high harmonics from metals is of general interest. To make this clear to a reader we have reworked the introduction to the paper to place the manuscript in a broader context than in the first draft. Specifically, we now discuss atomic and molecular optics, dielectrics and semiconductors, and plasma harmonic research. To us, metals fall neatly between solids and plasmas.

We have also taken all referee's criticisms to heart and made several changes.

- (a) We have added an analysis of the damage threshold of TiN to the text. We find that the damage threshold that we measure ($1.3 \times 10^{13} \text{ W/cm}^2$) is much below melting temperature of our 200 nm thick TiN sample.
- (b) To address the perturbative harmonic response of metals, we present a 1D model of harmonic emission from metals.
- (c) Most important, we have clarified our role as experimentalists, not theorists. Our experimental measurement of the angle dependence of the harmonics and how our measurement relates to the Fermi surface, show that an essential part of any theory will follow the motion of the average metal electrons. Pointing the direction for a full theory is what makes a good experiment.

Best regards,

A. Korobenko, S. Saha, A. T. K. Godfrey, M. Gertsvolf, A. Yu. Naumov, D. M. Villeneuve, A. Boltasseva, V. M. Shalaev, and P. B. Corkum

REVIEWERS' COMMENTS

Reviewer #1 (Remarks to the Author):

I would like to thank the authors for answering my questions and taking my comments into account, either by implementing changes in the manuscript or explaining their point of view whenever we disagree. I think the manuscript is more clear now.

However, I am still not convinced that this manuscript is fitted to Nature Communication, due to the lack of novelty and because the strong experimental requirements limit the spreading of HHG from metal to not specialized laser laboratories.

Moreover, I have two main comments.

1. In answer to one comment shared by the 3rd referee and myself, they modified the introduction to show that "observing high harmonics from metals is of general interest. To make this clear to a reader we have reworked the introduction to the paper to place the manuscript in a broader context than in the first draft. Specifically, we now discuss AMO, dielectrics and semiconductors, and plasma harmonic research." However, in my opinion the introduction is now difficult to follow, as the line of thought is going seemingly towards several different directions. This causes the main topic of the article to arrive late. This might deter the not specialized readers from going through the whole letter.

2. I am a bit confuse by the handling of the mathematical formulae. Indeed, they first present the final equation used to calculate the photo induced current (I228). Several paragraphs later, they introduce equation (1) and its origin, without linking it to equation I228, while it is the first step to derive the previous equation. In my opinion, the reverse order should be used and would be easier to follow.

Reviewer #2 (Remarks to the Author):

The revised manuscript has addressed my questions and concerns adequately. I believe that it will be of interest to the readers of NC since it should be the first experimental work on HHG in metals. I support the acceptance of his work after the authors do some minor revisions:

1)Page 2, "Fig. 2" should be "Fig. 1", and "Fig. 1" should be "Fig. 2".

2)Line 228 and the section "Current in a 1D metal" from Line 435 to 467, please refer to PRB 100, 214312 (2019), the results have been published.

3)The definition of the vector potential appears twice, see Lines 232 and 292.

4)Line 344, the authors propose to use nanoantenna with a dielectric to enhance HHG, please refer to [Nat. Commun. 7, 13105 doi: 10.1038/ncomms13105 (2016)] and [PRA 94, 023419 (2016)].

Reviewer #3 (Remarks to the Author):

I've read the authors' comments and the revised manuscript. Their responses are reasonable and appropriate. I agree that the fundamental process of HHG in TiN falls in between solids and plasmas. I thus agree the manuscript to be published in Nature Communications.

Please find below our reply to these comments:

Reviewer #1.

I would like to thank the authors for answering my questions and taking my comments into account, either by implementing changes in the manuscript or explaining their point of view whenever we disagree. I think the manuscript is more clear now.

However, I am still not convinced that this manuscript is fitted to Nature Communication, due to the lack of novelty and because the strong experimental requirements limit the spreading of HHG from metal to not specialized laser laboratories.

Moreover, I have two main comments.

1. In answer to one comment shared by the 3rd referee and myself, they modified the introduction to show that "observing high harmonics from metals is of general interest. To make this clear to a reader we have reworked the introduction to the paper to place the manuscript in a broader context than in the first draft. Specifically, we now discuss AMO, dielectrics and semiconductors, and plasma harmonic research." However, in my opinion the introduction is now difficult to follow, as the line of thought is going seemingly towards several different directions. This causes the main topic of the article to arrive late. This might deter the not specialized readers from going through the whole letter.

To address Reviewer #1's comment and to satisfy the Nature Communication formatting guide, we shortened the introduction section, and separated the discussion of the previous work from the summary of our results better. This made introduction easier to follow.

2. I am a bit confuse by the handling of the mathematical formulae. Indeed, they first present the final equation used to calculate the photo induced current (I228). Several paragraphs later, they introduce equation (1) and its origin, without linking it to equation I228, while it is the first step to derive the previous equation. In my opinion, the reverse order should be used and would be easier to follow.

Since the first equation referred to the 1D case, and the second to the 3D case, one was not intended to follow from the other. We agree that this could cause confusion. In the revised version, we modified the discussion of the 1D case, following the comment of the Reviewer #2, we replaced the derivation for the 1D case with a reference to the previously published paper. This makes the discussion of a 3D case more clear and should remove the confusion.

Reviewer #2 (Remarks to the Author):

The revised manuscript has addressed my questions and concerns adequately. I believe that it will be of interest to the readers of NC since it should be the first experimental work on HHG in metals. I support the acceptance of his work after the authors do some minor revisions:

1)Page 2, "Fig. 2" should be "Fig. 1", and "Fig. 1" should be "Fig. 2".

Fixed the numbering.

2)Line 228 and the section “Current in a 1D metal” from Line 435 to 467, please refer to PRB 100, 214312 (2019), the results have been published.

We were not aware of this paper. Since essentially identical series expansion of the moving electron energy, we decided to replace our derivation from “Current in a 1D metal” subsection of Methods and cite the suggested reference instead. The main text was edited accordingly.

3)The definition of the vector potential appears twice, see Lines 232 and 292.

Fixed the double definition.

4)Line 344, the authors propose to use nanoantenna with a dielectric to enhance HHG, please refer to [Nat. Commun. 7, 13105 doi: 10.1038/ncomms13105 (2016)] and [PRA 94, 023419 (2016)].

The suggested citations were added.

Reviewer #3 (Remarks to the Author):

I've read the authors' comments and the revised manuscript. Their responses are reasonable and appropriate. I agree that the fundamental process of HHG in TiN falls in between solids and plasmas. I thus agree the manuscript to be published in Nature Communications.

Best regards,

A. Korobenko, S. Saha, A. T. K. Godfrey, M. Gertsyovf, A. Yu. Naumov, D. M. Villeneuve, A. Boltasseva, V. M. Shalaev, and P. B. Corkum